# Theoretical Design of a Janus-Nanoparticle-Based Sandwich Assay for Nucleic Acids

**DOI:** 10.3390/ijms23158807

**Published:** 2022-08-08

**Authors:** Takumi Sato, Keiko Esashika, Eiji Yamamoto, Toshiharu Saiki, Noriyoshi Arai

**Affiliations:** 1Department of Mechanical Engineering, Keio University, Kohoku-ku, Yokohama 223-8522, Japan; 2Department of Electrical and Information Engineering, Keio University, Kohoku-ku, Yokohama 223-8522, Japan; 3Department of System Design Engineering, Keio University, Kohoku-ku, Yokohama 223-8522, Japan

**Keywords:** nanosensor, molecular simulation, nanoparticle, self-assembly, surface-enhanced Raman scattering

## Abstract

Nanoparticles exhibit diverse self-assembly attributes and are expected to be applicable under unique settings. For instance, biomolecules can be sandwiched between dimer nanoparticles and detected by surface-enhanced Raman scattering. Controlling the gap between extremely close dimers and stably capturing the target molecule in the gap are crucial aspects of this strategy. Therefore, polymer-tethered nanoparticles (PTNPs), which show promise as high-performance materials that exhibit the attractive features of both NPs and polymers, were targeted in this study to achieve stable biomolecule sensing. Using coarse-grained molecular dynamics simulations, the dependence of the PTNP interactions on the length of the grafted polymer, graft density, and coverage ratio of a hydrophobic tether were examined. The results indicated that the smaller the tether length and graft density, the smaller was the distance between the PTNP surfaces ($R_{\mathrm{surf}}$). In contrast, $R_{\mathrm{surf}}$ decreased as the coverage ratio of the hydrophobic surface (ϕ) increased. The sandwiching probability of the sensing target increased in proportion to the coverage ratio. At high ϕ values, the PTNPs aggregated into three or more particles, which hindered their sensing attributes. These results provide fundamental insight into the sensing applications of NPs and demonstrate the usefulness of PTNPs in sensing biomolecules.

## 1. Introduction

Nanosensors have a wide range of applications in the chemical, optical, biomedical, food, and electronics industries [1,2,3]. They can detect chemical or mechanical activity and monitor physical quantities such as temperature at the molecular scale [4,5,6]. Essentially, nanosensors are extremely beneficial devices that can convert microscopic information derived from the behavior of atoms and molecules into macroscopic data [7,8,9]. In particular, nanosensors based on the surface-enhanced Raman scattering (SERS) effect of metal nanoparticles (NPs) [10,11,12,13] have attracted considerable attention. In the design of these nanosensors, controlling the interparticle distance of the NPs is crucial because of its significant impact on SERS [14,15,16,17]. DNA sensing is a particularly encouraging application of this detection strategy as it can be applied, for instance, in cancer diagnosis for microRNA identification; however, regulating the position of the measurement target is challenging. Experimental and simulation studies have been conducted to control the gap by decorating NP surfaces. Esashika et al. developed a method to dimerize gold NPs by exploiting the van der Waals interactions between alkyl chains grafted onto the NP surfaces [18]. Ye et al. showed that the use of polymeric ligands in the polymer component can achieve nanometer-level control of the interparticle distances in complex 3D structures [19]. In this regard, Raman-based sequencing can be realized in the plasmonic gap mode if a mechanism can be established to shed DNA molecules into this gap. Therefore, the present simulation-based study was designed to theoretically investigate the possibility of achieving stable sensing using NPs whose surfaces were grafted with polymers. The NPs grafted with polymers—also known as polymer-tethered NPs (PTNPs)—are expected to be high-performance materials that combine the advantages of NPs (optical, electrical, and magnetic properties) with those of polymers (high flexibility and processability). The PTNPs have two characteristic lengths that can significantly affect their functionality: a hard short length corresponding to the radius of the NP and a soft long length signified by the radius of the polymer [20,21].

Various self-assembled structures of NPs have been revealed by simulations [22,23,24,25] and experiments [26]. For example, Janus NPs—also known as patchy NPs—are representative modified NP systems that exhibit unprecedented unique self-assembled structures. Essentially, Janus NPs are NPs that exhibit two different chemical properties on their NP surfaces. Depending on their surface properties and Janus balance, Janus NPs exhibiting diverse self-assembling behavior, such as bilayers and hexagonal crystals, have been realized [27,28,29,30]. Additionally, advances in microfabrication have enabled the preparation of nanoparticles with physically and chemically complex surfaces and the creation of morphologies with diverse properties. Moreover, the degree of polymerization and graft density of the grafted polymers in PTNPs have been found to affect the interparticle distance of the NPs [31]. Through simulations, Dukes et al. revealed that the morphology of the grafted surface varies with the length and density of the polymer [32]. However, self-assembled structures of macromolecule–PTNP mixtures have not yet been investigated for the sensing of macromolecules such as DNA.

In this study, coarse-grained molecular dynamics simulations were performed to examine the interactions between Janus-type tethered NPs and biomolecules of interest. The length, density, and properties of the grafted polymers, as well as the Janus balance were varied in this study to provide fundamental insight into the sensing applications of PTNPs.

## 2. Results and Discussions

PTNPs suitable for molecular sensors were designed and investigated in this study using simulations.

Information regarding molecules that are near the interface of two PTNPs can be obtained by applying an electric field between them. Therefore, to impart sensing characteristics to the investigated PTNP system, the two NPs had to be stably maintained in close proximity for trapping a target molecule between their surfaces. The grafting of polymers onto the NP surfaces is expected to permit energy-efficient pinning of target molecules via self-assembly phenomena.

Overall, the study was conducted in two parts. First, the dependence of the distance between the PTNPs on various factors was investigated to optimize their design. Second, the PTNPs in the optimized configuration were examined to determine whether they could adequately sandwich the target molecule and permit stable sensing.

### 2.1. Design of PTNPs Suitable for Sensing

Three parameters—the tethered-polymer length (*L*), graft density (ρ), and Janus balance (ϕ), which is the coverage ratio of the hydrophobic surface—were targeted to examine the influence of the PTNP design on sensing (Figure 1). Simulations were performed to analyze the interactions between two PTNPs, as shown in Figure 2. Figure 3 shows the variations in the distance between the PTNP surfaces ($R_{\mathrm{surf}}$) with ϕ for different values of ρ and *L*.

First, for all *L* values, lower ρ values led to lower $R_{\mathrm{surf}}$ values. This is because at high ρ values, the layer formed by the tethered polymer cannot penetrate the other nanoparticle, as shown in Figure 3d.

With respect to ϕ, $R_{\mathrm{surf}}$ was observed to increase as ϕ decreased. At low ϕ values, the hydrophobic tethered polymers aggregated and stacked in the radial direction owing to the inhibitive effect of the surrounding hydrophilic polymers. Therefore, the hydrophobic tethered polymer layer became thick and $R_{\mathrm{surf}}$ increased, in contrast to the scenario with a larger ϕ (Figure 3e).

In terms of the length *L*, $R_{\mathrm{surf}}$ increased as *L* increased in the range of L≤6 because, as shown in Figure 3f, the thickness of the hydrophobic layer formed on the nanoparticle surface increased as the length of the tethered polymer increased. Esashiya et al. [18] obtained a similar result by calculating the variation in the distance between PTNP surfaces ($R_{\mathrm{surf}}$) with the alkyl chain length (hydrophobic polymer length) using the finite-difference time-domain method based on extinction spectra. $R_{\mathrm{surf}}$ was found to increase in proportion to the alkyl chain length. Although Esashika et al. used a sufficiently small graft density to prevent alkyl chain entanglement, the graft density examined in this study also showed an increase in $R_{m}athrmsurf$ with increasing *L*.

Specific results will now be discussed in detail. The degree of influence of ϕ on $R_{\mathrm{surf}}$ depended on *L*. For example, at the lowest investigated *L* value (3), $R_{\mathrm{surf}}$ changed slightly for ϕ≤20. However, at high *L* (*L* = 9), $R_{\mathrm{surf}}$ decreased as ϕ increased across its entire range. Figure 4a–c show representative snapshots of PTNPs at equilibrium, acquired at ρ=6.0, ϕ=50%, and different lengths. For *L* = 3, only the hydrophobic polymers close to their nanoparticle surfaces formed a contact layer, whereas for *L* = 9, all hydrophobic polymers formed a cylindrical contact layer. Thus, because of the variation in the contact layer with *L*, the dependence of ϕ on $R_{\mathrm{surf}}$ also varied with *L*.

In the case of *L* = 9, compared with *L* = 6, $R_{\mathrm{surf}}$ decreased, despite the high *L* value under all conditions except for ρ = 0.8 (Figure 3c). This is because of the fact that for a lengthy tethered polymer (Figure 3d), bridges of the hydrophobic tethered polymer are formed between the PTNPs, leading to strong attraction between the PTNPs. As shown in Figure 4c (*L* = 9), all hydrophobic polymers formed a contact layer regardless of ϕ. The formation of a larger bridged area than that in the *L* = 6 scenario led to stronger attraction between the PTNPs, resulting in a decrease in $R_{\mathrm{surf}}$.

Interestingly, trends different from those of the other configurations were observed at low graft densities (Figure 3a–c). In particular, for ρ = 0.8, $R_{\mathrm{surf}}$ increased at high coverage ratios (ϕ∼ 35–50%), whereas it decreased with ϕ for the other ρ values. Figure 4d–h show snapshots of PTNPs with the left side at equilibrium for different ϕ values. At high coverage ratios (ϕ≥40%, Figure 4d,e), the hydrophobic polymers formed multiple aggregated domains on the nanoparticle surface. These domains caused local stacking upon contact with the other PTNP, which led to an increase in $R_{\mathrm{surf}}$. This local stacking is likely difficult to control, particularly because irregular variations in $R_{\mathrm{surf}}$ were observed for ϕ≥ 30% at *L* = 9.

After investigating the effects of *L*, ϕ, and ρ on $R_{\mathrm{surf}}$, the sensing applications of the system were targeted. In this regard, effective sensing was considered to occur at $R_{\mathrm{surf}}$ values of less than 1 nm. Therefore, the configuration with *L* = 3 and ρ = 0.8 was adopted because an $R_{\mathrm{surf}}$ value of less than 1 nm could be easily achieved. This configuration was further investigated to determine the possibility of the target molecule being appropriately sandwiched.

### 2.2. Simulations for Biomolecule Sensing

The capability of the PTNPs in the aforementioned optimized configuration to adequately interlace a target molecule and thereby achieve stable sensing was subsequently examined. To this end, simulations were started from the initial state shown in Figure 2b to achieve a “sandwiched state”, which was considered to occur when the distance between the target molecule and the two nanoparticle surfaces was within 1 nm during the 0.1 μs simulation. Figure 5b shows a snapshot of a typical sandwiched state. One hundred simulations were performed under each of the following conditions to determine the extent to which the sandwiched state was achieved: L=3, ρ = 0.8, and ϕ = 10–50.

Figure 5 shows the sandwiching probability ($P_{\mathrm{sand}}$) corresponding to each ϕ value. $P_{\mathrm{sand}}$ was highest (82%) at ϕ=40% and lowest (6%) at ϕ=10%. Our model assumes that the PTNPs achieve sensing by trapping a target molecule in their hydrophobic polymer region. In other words, the probability that the target molecule does not contact the hydrophobic polymer region increases, and $P_{\mathrm{sand}}$ decreases.

However, the $P_{\mathrm{sand}}$ corresponding to ϕ = 50%, which has the widest hydrophobic region, was lower than that corresponding to ϕ = 40%. This is due to the capture of the target molecule beyond the measurement area, as shown in Figure 5d,e. As mentioned above, when ρ is low and ϕ is high, the hydrophobic polymer forms multiple aggregated domains and becomes trapped in one of them. This inappropriate trapping is more likely to occur when ϕ is high; this was confirmed in the simulations conducted with ϕ≥ 40%. $P_{\mathrm{sand}}$ was found to increase as ϕ increased; however, the probability that the target molecule was trapped on only one side of the NP surfaces increased. Therefore, ϕ and $P_{\mathrm{sand}}$ had a trade-off relationship.

To date, simulation-based studies have typically been performed on isolated systems with only two PTNPs and a target molecule. Because SERS is an effect that occurs between nanoparticles, it can be measured even for trimers or higher oligomers. Typically, dimers are preferred for general use because they have a stable structure and are limited to a single position. However, multiple PTNPs must be used to realize effective PTNP-based sensing systems. Nevertheless, the stable creation and separation of dimers is a challenge in sensor development because of the aggregating tendency of gold NPs. Therefore, simulations of aqueous solutions containing a large number of PTNPs (twelve) in the system were performed for each ϕ to determine the aggregation number at equilibrium. Three or more aggregations were observed to be formed at ϕ≥30%, as shown in Figure 6. Furthermore, when ϕ=10%, no dimer formation occurred, and stable dimer formation was possible at ϕ=20 (Figure 6c). This result along with the observations from Figure 5 indicate that the design with ϕ=20 led to the most stable sensing behavior.

## 3. Materials and Methods

### 3.1. Dissipative Particle Dynamics Simulations

The dissipative particle dynamics (DPD) method [33,34], which is a powerful technique that enables the analysis of phenomena occurring on *millisecond* time scales and *micrometer* length scales, was implemented in this study using in-house code. Each DPD particle represented one coarse-grained bead that was composed of a group of atoms or molecules. The DPD method has been applied to examine the attributes of various NP systems, such as interactions between NPs and a lipid bilayer [35,36], the thermal conductivity of nanofluids [37,38], and the self-assembly of PTNPs [22,39].

In this method, each DPD particle obeys Newton’s equation of motion, with three types of forces acting on all beads: $F^{C}$, $F^{R}$, and $F^{D}$, which represent conservative, pairwise random, and dissipative forces, respectively.
(1)$$m_{i}\frac{dv_{i}}{dt}=f_{i}=\sum_{j\neq i}F_{ij}^{C}+\sum_{j\neq i}F_{ij}^{D}+\sum_{j\neq i}F_{ij}^{R}\;\;$$
where *m* is the mass and v is the velocity. The conservative force $F^{C}$ is expressed as follows:(2)$$F_{ij}^{C}=\left\{\begin{array}{cc} -a_{ij}\left(1-\frac{\left|r_{ij}\right|}{r_{c}}\right)n_{ij}, & \left|r_{ij}\right|\leq r_{c}\\ \;\;\;\;\;\;\;\;\;\;\;\;\;\;\;0, & \left|r_{ij}\right|>r_{c}\;\;, \end{array}\right.$$
where $r_{ij}=r_{j}-r_{i}$, $n_{ij}=r_{ij}/\left|r_{ij}\right|$, **r** is the position vector of a particle, $a_{ij}$ is a parameter that determines the magnitude of the repulsive force between particles *i* and *j*, and $r_{c}$ is the cutoff distance for determining the effective range of the force. The random ($F_{ij}^{R}$) and dissipative forces ($F_{ij}^{D}$) are expressed as follows:(3)$$F_{ij}^{R}=\left\{\begin{array}{cc} \rho \omega^{R}\left(\left|r_{ij}\right|\right)\zeta_{ij}\Delta t^{-1/2}n_{ij}, & \left|r_{ij}\right|\leq r_{c}\\ \;\;\;\;\;\;\;\;\;\;\;\;\;\;\;0, & \left|r_{ij}\right|>r_{c} \end{array}\right.$$
and
(4)$$F_{ij}^{D}=\left\{\begin{array}{cc} -\gamma \omega^{D}\left(\left|r_{ij}\right|\right)\left(n_{ij}\cdot v_{ij}\right)n_{ij}, & \left|r_{ij}\right|\leq r_{c},\\ \;\;\;\;\;\;\;\;\;\;\;\;\;\;\;0, & \left|r_{ij}\right|>r_{c} \end{array}\right.$$
where $v_{ij}=v_{j}-v_{i}$, ρ denotes the noise parameter, γ denotes the friction parameter, and $\zeta_{ij}$ is a random number based on the Gaussian distribution. Here, $\omega^{R}$ and $\omega^{D}$ are *r*-dependent weight functions, expressed as
(5)$$\omega^{D}\left(r\right)={\left[\omega^{R}\left(r\right)\right]}^{2}=\left\{\begin{array}{cc} {\left[1-\frac{\left|r_{ij}\right|}{r_{c}}\right]}^{2}, & r_{ij}\leq r_{c}\\ \;\;\;\;0, & r_{ij}>r_{c}\;\;. \end{array}\right.$$

The temperature is controlled by coupling the dissipative and random forces, whereas ρ and γ are interconnected by the fluctuation–dissipation theorem.
(6)$$\rho^{2}=2\gamma k_{B}T,$$
where $k_{B}$ is the Boltzmann constant and *T* is the temperature.

Reduced units are generally used in DPD simulations. Here, the units of length, mass, and energy are the cutoff distance $r_{c}$, the bead mass *m*, and $k_{B}T$, respectively. The method established by Groot and Rabone was adopted to scale the length and time units [40]. This method considers the number of water molecules in a single DPD particle and the diffusion constant of water.

### 3.2. Simulation Models and System

The PTNP model used in this study is illustrated in Figure 1. Each NP was treated as a rigid body [41,42], and its radius was set to 3.9 nm. Moreover, each NP was composed of 18,168 DPD beads on a diamond lattice with a lattice constant of 0.47 nm. The PTNP models were constructed by varying the tethered polymer length *L*, graft density ρ, and coverage ratio of the hydrophobic surface ϕ. The graft density is defined as follows:(7)$$\rho =\frac{N_{g}}{\pi D^{2}},$$
where $N_{g}$ is the number of grafted chains and *D* is the diameter of the NP.

The graft polymers considered in this study comprised two types of chemicals based on their affinity to a solvent: hydrophilic and hydrophobic (denoted as HI and HO, respectively). A harmonic spring force ($F_{ij}^{S}$) was introduced to reproduce the bond between the polymers. The spring force is expressed as $F_{ij}^{S}=-k_{s}\left(\left|r_{ij}\right|-r_{s}\right)n_{ij}$, where $k_{s}$ and $r_{s}$ are the spring constant and the equilibrium bond length, which were set as 100 $k_{B}T/r_{c}^{2}$ and 0.56 nm, respectively. The solvent (denoted S) was treated as a single bead, and the measurement target was considered to be 20 beads long (Figure 1d).

As mentioned earlier, PTNPs with different graft densities, tether lengths, and coverages were investigated. Each graft polymer was linear and comprised three, six, or nine DPD beads (Figure 1a1–a3). Graft densities of 0.8, 1.5, 3.0, and 6.0 were used (Figure 1b1–b4), in addition to coverage ratios of 10–50% (Figure 1c1–c5). The interaction parameters of the DPD beads are listed in Table 1; the P, I, O, D, and S notations correspond to beads of the nanoparticles, hydrophilic polymer, hydrophobic polymer, target polymer, and solvent, respectively. The interaction between two particles is dictated by $a_{ij}$, whose value is determined by the solubility parameter. Various values have been considered for this interaction parameter in previous studies; in most cases, the water–hydrophilic-part and water–hydrophobic-part interactions exhibit values in the range of 25 and 50–130, respectively [22]. In this study, the value of $a_{ij}$ was modified in accordance with the measurement target; however, representative parameters were described because the overall trend was consistent. In this study, the target molecules were assumed to be hydrophobic, but many biomolecules exhibit hydrophilic properties. To sense hydrophilic molecules by the mechanism provided in this study, hydrophobic solvents such as oil, for example, can be used.

Two PTNPs and solvent beads were included in the simulation supercell. The diameter of the NPs was set to 7.8 nm.

In the initial configuration of the simulation system (Figure 2), the PTNPs were initially fixed, and periodic boundary conditions were applied in all directions. All the simulations were performed in a constant-volume and constant-temperature ensemble. The volume of the simulation box was 25.8 × 19.4 × 19.4 nm${}^{3}$, and the simulations were conducted at room temperature (1.0 $k_{B}T$).

## 4. Conclusions

The DPD method was used in this study to investigate the surface-modification-induced changes in the distance between Janus-type tethered NPs ($R_{\mathrm{surf}}$), as well as the sandwiched state. Three different tether lengths (*L*), three different graft densities (ρ), and five different coverage ratios (ϕ) of the Janus-type tethered NPs were considered.

The distance between the NP surfaces, which was estimated for each model, was found to increase in proportion to the tether length and graft density. Moreover, the larger the coverage, the smaller was the distance between the NP surfaces. Exceptions to these trends were observed at low graft densities. The PTNP design with *L* = 3 and ρ = 0.8 was selected for biomolecule sensing applications because an $R_{\mathrm{surf}}$ value of less than 1 nm could be readily obtained.

The target-molecule-sandwiching probability at *L* = 3 and ρ = 0.8 increased with increasing coverage ratio. However, at ϕ≥30%, aggregations of more than three PTNPs were observed, making them unsuitable for sensing applications. Thus, the PTNP design with ϕ=20 is expected to display the most stable sensing behavior.

DNA sensing demonstrates promising applications as a detection strategy, for instance in cancer diagnosis for microRNA identification; however, controlling the position of the measurement target remains difficult. The findings of this study may assist in providing fundamental insight into the sensing applications of NPs and enabling facile detection of DNA.

## Figures and Tables

**Figure 1 ijms-23-08807-f001:**
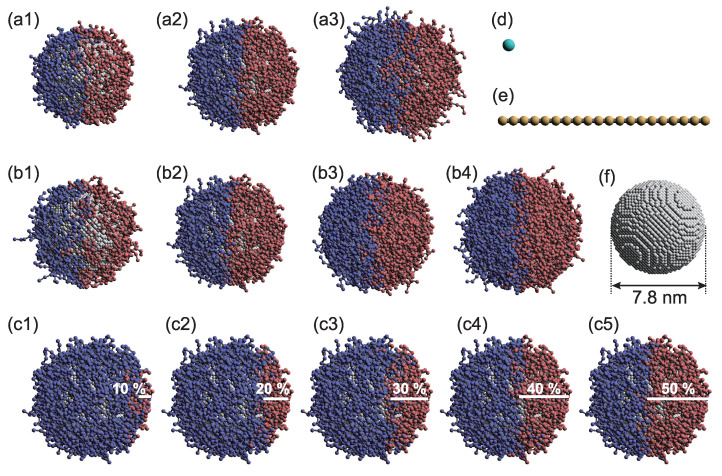
Simulation models. Variations in the (**a1**–**a3**) tethered polymer length *L* ((**a1**) three, (**a2**) six, and (**a3**) nine beads); (**b1**–**b4**) graft density ρ ((**b1**) 0.8, (**b2**) 1.5, (**b3**) 3.0, and (**b4**) 6.0); and (**c1**–**c5**) coverage ratio ϕ ((**c1**) 10%, (**c2**) 20%, (**c3**) 30%, (**c4**) 40 %, and (**c5**) 50%) of polymer-tethered nanoparticles (PTNPs) with Janus surfaces. Models of a (**d**) solvent molecule, (**e**) 20-bead-long target polymer, and (**f**) nanoparticle. Color code: nanoparticles, grey; hydrophilic groups, blue; hydrophobic groups, red; and measurement target, yellow.

**Figure 2 ijms-23-08807-f002:**
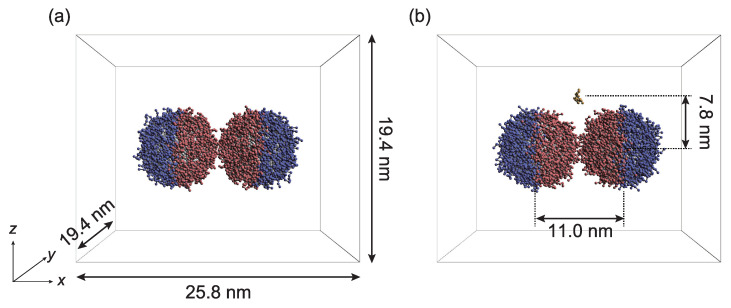
Initial configurations of PTNPs for simulations performed to (**a**) optimize their design and (**b**) determine their sensing capability. Solvent beads are not shown for clarity.

**Figure 3 ijms-23-08807-f003:**
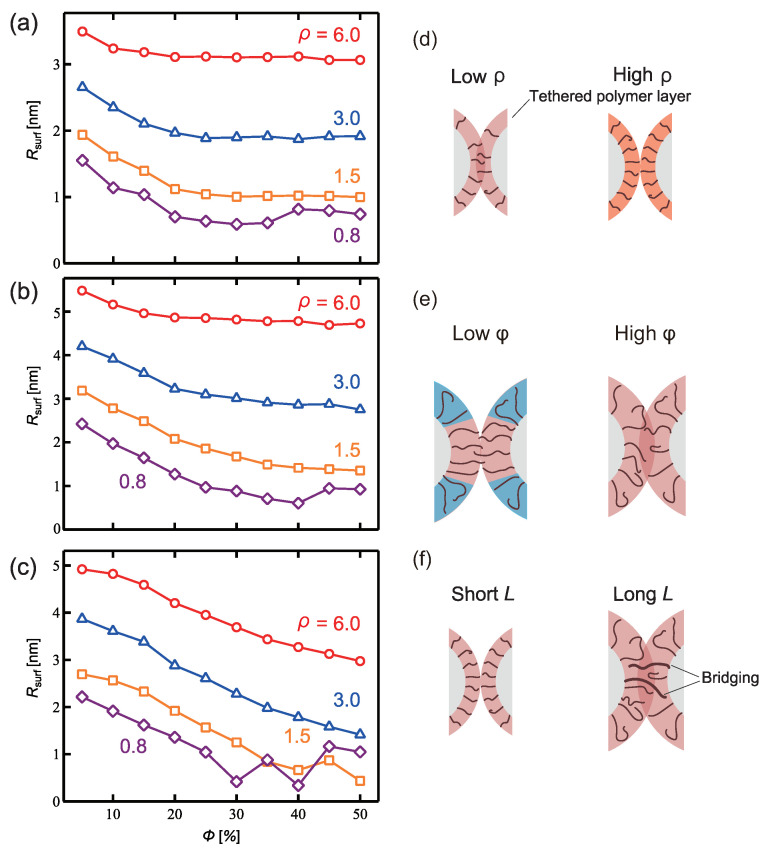
Distance between PTNP surfaces ($R_{\mathrm{surf}}$) with respect to the coverage ratio ϕ and tethered polymer length *L*: (**a**) *L* = 3, (**b**) *L* = 6, and (**c**) *L* = 9. Schematics illustrating the differences in $R_{\mathrm{surf}}$ according to (**d**) ρ, (**e**) ϕ of a hydrophobic tether, and (**f**) *L*.

**Figure 4 ijms-23-08807-f004:**
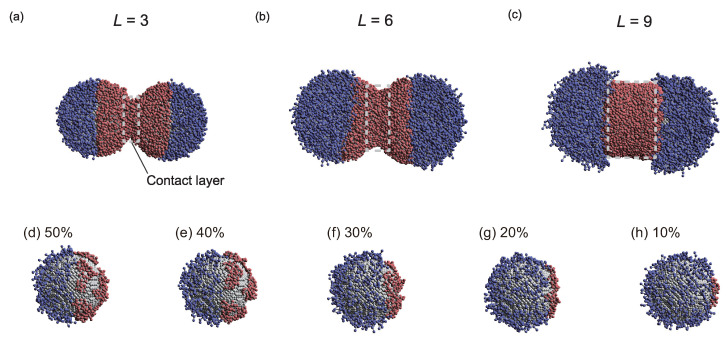
Snapshots of PTNPs at equilibrium acquired at ρ = 6.0 and ϕ = 50%: (**a**) L=3, (**b**) L=6, and (**c**) L=9. Snapshots of the PTNPs with *L* = 3 and their left side at equilibrium for ϕ values of (**d**) 50%, (**e**) 40%, (**f**) 30%, (**g**) 20%, and (**h**) 10%.

**Figure 5 ijms-23-08807-f005:**
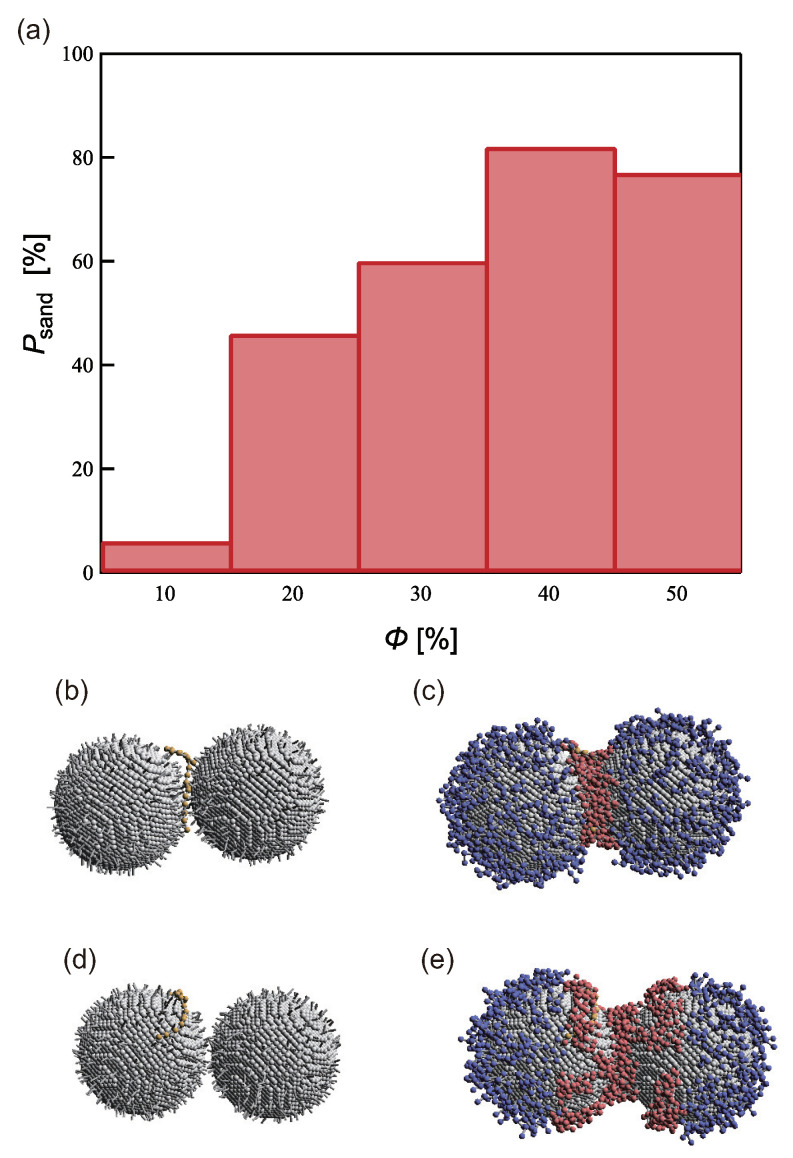
(**a**) Sandwiching probability of sensing the target at each coverage ratio for *L* = 3 and ρ = 0.8. (**b**,**d**) Snapshots of the PTNPs illustrating (**d**) effective and (**e**) ineffective (ϕ = 50%) capture of the sensing target; (**c**,**e**) show the same snapshots with the grafted polymer particles, which were hidden in (**b**,**d**) for clarity.

**Figure 6 ijms-23-08807-f006:**
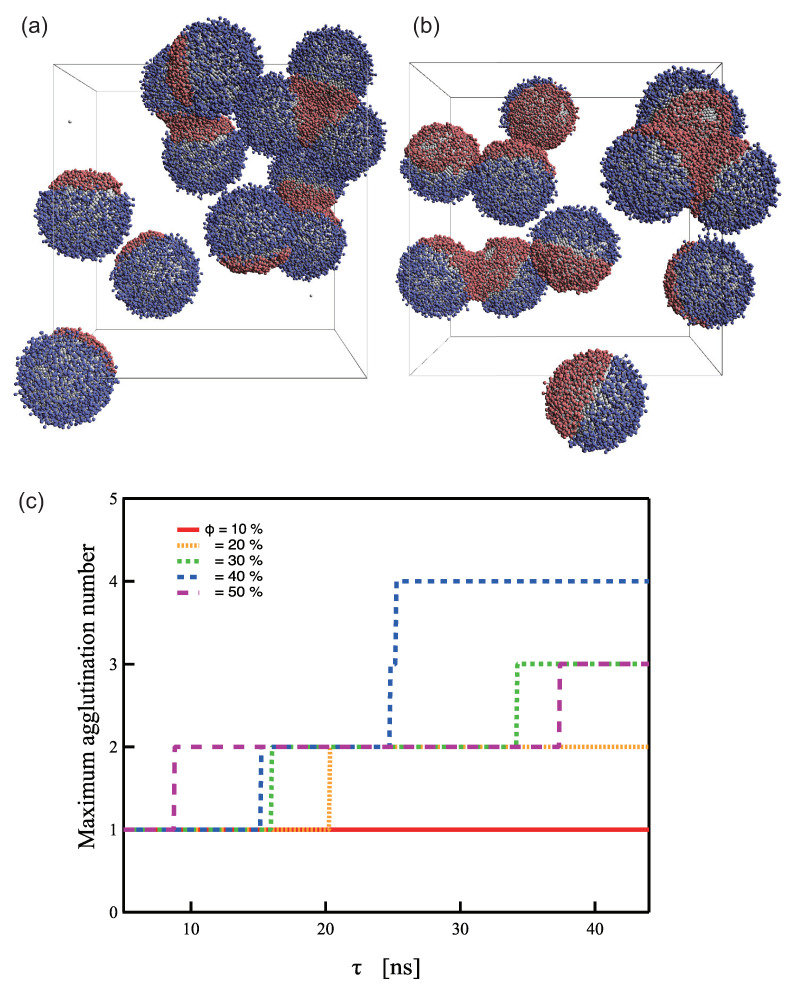
Representative snapshots of PTNPs in equilibrium at ϕ = (**a**) 30% (**b**) 50% and (**c**) Maximum agglutination number at each coverage ratio.

**Table 1 ijms-23-08807-t001:** Interaction parameters of the beads $a_{ij}$ (unit: $k_{B}T/r_{c}$).

	P	I	O	D	S
P	25	50	100	25	25
I	50	25	100	100	25
O	100	100	25	25	100
D	25	100	25	25	75
S	25	25	100	75	25

## Data Availability

Not applicable.

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
