# Peer review of "Theoretical Design of a Janus-Nanoparticle-Based Sandwich Assay for Nucleic Acids"

_ijms, 2022, doi:10.3390/ijms23158807_

Round 1

Reviewer 1 Report

A brief summary:

Sato et al. reported a theoretical/computational study of designing Janus-type polymer-tethered nanoparticle (PTNP) sensors for general target molecules. Using coarse-grained dissipative particle dynamics simulations, this work investigated how tethered polymer length and graft density of PTNP regulate the inter-molecular distance between two nanoparticles, which should be in sufficient proximity for sensing. The authors then simulated a target molecule with two PTNPs and found an optimal hydrophobic coverage ratio that can keep the target sandwiched by the PTNPs without their self-aggregation. Overall, this study provides insights into selecting key parameters for effective PTNP sensors that can be applied in further experimental nanosensor design.

The main theme and some results of this manuscript, in general, meet the scope and standard of the International Journal of Molecular Sciences, especially in the section on Physical Chemistry and Chemical Physics and the Special Issue “2nd Edition: Advances in Molecular Simulation”. However, some key parts of the method and results were not explained well enough to support the conclusion of “designing nanosensors for proteins and nucleic acids”. Also, the motivation of the entire study was not well connected with the actual research content. Therefore, I recommend reconsideration of this manuscript, after the major concerns and other minor issues (detailed below) are adequately addressed in a revised version.

General concept comments:

1.    The title and introduction are written in a way that suggests the major motivation of the study to provide “fundamental insights” into specific experimental applications of nanosensors, including Surface-Enhanced Raman Scattering (SERS) and DNA sensing. However, the subsequent research content turns out to be a very general computational study without any experimental validation. To improve the scientific soundness of the manuscript, it would be ideal to perform at least one experiment to test the nanosensor designed by computer simulations. If it is not a feasible task for the authors, then the motivation should be rewritten to emphasize the sensing mechanism of PTNPs from a physical/chemical perspective and discuss the key attributes that leverage their sensing efficiency.

2.    The entire study is based on dissipative particle dynamics (DPD) simulations. Therefore, appropriate parameters of the DPD force field are crucial for the reliability of the results and conclusion. Unfortunately, reasonable parameterization procedure or reference of some key parameters, especially the interaction parameters of the beads aij listed in Table 1, are completely missing. Comprehensive reasoning of aij selection will dramatically help readers understand the physical properties of the system (i.e., the type of nanoparticle beads, sensor target polymer beads, and solvent) simulated in this work. For example, if the parameters of the target molecule represent similar properties of nucleic acids, then the results might bring insights for DNA sensor design.

3.    The authors also performed simulations of multiple PTNPs to investigate the aggregation behavior and determined the optimal coverage ratio before aggregation happens. However, the fundamental reason why aggregation can necessarily impair sensing efficiency in practical applications, such as SERS and DNA sensing, was not adequately explained with relevant references. Also, target molecules should be included in the aggregation simulations, so that their “sandwich state” can be directly measured in a more realistic crowded environment.    

Specific comments:

·  Page 3 line 85: coverage ratio should have a clear definition at its first appearance.

·  Figure 3(a-c) and Figure 5(a): since they are results of multiple frames/simulation runs, error bars should be included.

·  Page 8 line 203-205: aggregation results of phi = 10% and 20% should be described and compared with other phi values.

·  Figure 6: instead of only showing representative snapshots, the authors should define a metric to quantify “the level of aggregation” and use it to distinguish systems with different phi values.

Reviewer 2 Report

Manuscript ID: ijms-1813141

Title: Theoretical Design of a Janus-Nanoparticle-Based Sandwich Assay for Proteins and Nucleic Acids 

This manuscript presents a theoretical study to obtain a stable sensor using polymer-tethered nanoparticles (PTNPs), as high-performance materials owing to their attractive features that are acquired from both NPs and polymers. Using coarse-grained molecular dynamics simulations, the dependence of the PTNP interactions on the length of the grafted polymer, graft density, and coverage ratio of a hydrophobic tether were examined.

The manuscript is well written, and the simulations are well designed, but some aspects of the paper need to be improved.

Abstract.

The authors stated: "These results provide fundamental insight into the sensing applications of NPs and demonstrate the usefulness of PTNPs in sensing biopolymers.". Please clarify if you want to use PTNPs in the detection of biopolymers or biomolecules.

1.Introduction

Please provide some information about Janus particles (short description, explanation).

The title refers at Nucleic Acids and proteins. The state of art is referring just at DNA. Please add more information about protein sensing.

2. Materials and Methods

2.1. Dissipative particle dynamics (DPD) simulations

Please provide the significance of ri and rj.

2.2. Simulation models and system

Please provide information about the software used by the authors for simulations and calculations.

4. Conclusions

Authors stated "The findings of this study may assist in providing fundamental insight into the sensing applications of NPs and enabling facile detection of DNA. " Please explain this statement in a way that justifies the presence of the word Nucleic Acids in the title.

I didn't find any claims about proteins. If you do not add information to justify this, I think it would be good to remove the word "proteins" from the title.

Reviewer 3 Report

The paper of Sato et al. deals with an interesting issue of the development of molecular sensors. As the title of the work reveals, it is an exclusively theoretical work, and although it is written well and relatively simple, it also offers a precise and accurate discussion. However, I miss some experimental data. Much more, the authors introduced quite deeply Raman spectroscopy and metal nanoparticles as a field with high potential for nanosensor developments. Thus, I recommend testing the theoretically optimized PTNPs (polymer-tethered nanoparticles) in an experiment, i.e. to carry out preliminary SERS measurements of selected target molecules using the proposed SERS substrates = optimized PTNPs.

Regarding the formal part of the manuscript, I have just a few comments:

- Please, check the English language of the text. For example, pg. 2, ln.44-47: "In this method, NP dispersions could be used to capture DNA, possibly enabling sensing that is free from the issues of nanopore sequencers, which is the difficulty of capturing long molecular chains a difficult task."

- Pg. 4, ln. 100-101: "Because each NP was constructed using 18,168 beads, the total number of beads was 273,895." I am sorry, but I could not understand the counts. Is it important to know the exact number(s)?

-  Regarding the simulations for biomolecule sensing, how well does a target molecule defined in this study reflect molecules of proteins or nucleic acids?

Round 2

Reviewer 1 Report

In this revised manuscript, the authors adequately addressed all my concerns and comments on the previous version. The current manuscript has met the standard of IJMS and I recommend acceptance of it, with a further change mentioned below.

Fig. R2 in the authors' response letter accurately measures the agglomeration behavior of the simulated multiple PTNP systems. It should be added as a panel of Fig. 6 in the final version of this manuscript.